# Floquet Spectral Almost-Periodic Modulation of Massive Finite and Infinite Strongly Coupled Arrays: Dense-Massive-MIMO, Intelligent-Surfaces, 5G, and 6G Applications

**Hamdi Bilel *** 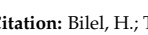 **and Aguili Taoufik**

Communication System Laboratory Sys'Com, National Engineering School of Tunis (ENIT),
University of Tunis El Manar, BP 37, Le Belvédère, Tunis 1002, Tunisia; taoufik.aguili@gmail.com
* Correspondence: hbilel.enit@gmail.com; Tel.: +32-466422407

**Abstract:** In this study, we introduce a new formulation based on Floquet (Fourier) spectral analysis combined with a spectral modulation technique (and its spatial form) to study strongly coupled sublattices predefined in the infinite and large finite extent of almost-periodic antenna arrays (e.g., metasurfaces). This analysis is very relevant for dense-massive-MIMO, intelligent-surfaces, 5G, and 6G applications (used for very small areas with a large number of elements such as millimeter and terahertz waves applications). The numerical method that is adopted to model the structure is the method of moments simplified by equivalent circuits MoM GEC. Other numerical methods (such as the ASM-array scanning method and the windowing Fourier method) used this analysis in their kernel to treat periodic and pseudo-periodic (or quasi-periodic) arrays.

**Keywords:** Floquet analysis; MoM method; almost-periodic antenna arrays; Fourier analysis; strong mutual coupling; dense massive MIMO; mm and THz waves; 5G and 6G applications

## 1. Introduction

Antenna arrays, and in particular dense (or massive) coupled almost-periodic antenna arrays, have been of great interest in telecommunications and RF electronic applications (such as dense-massive-MIMO, smart-surfaces, 5G, and 6G applications) [1–4], including those used for very small surfaces with large numbers of elements such as millimeter and terahertz array applications. Therefore, the spectrum analysis based on a Fourier transformation (in the Floquet domain) is proposed to simplify the EM calculation on an elementary cell surrounded by periodic walls, as explained in [1–7] (in other research, they use periodic Green's functions) [8–19]. In the bibliography and recent studies, only spatial modulation techniques have been proposed to study periodic systems with large sizes [20–23]. Except in our case, a Fourier spectral analysis is presented to introduce a spectral modulation technique and its spatial equivalent (Fourier and Fourier inverse) to study strongly coupled sub arrays in an infinite and large finite almost-periodic support [24–27]. In this context, several numerical methods such as FDTD and FEM and other integral methods like the method of moments and full-wave methods [28] are proposed to resolve the given problem. In our work, we are interested only in the method of moments combined with equivalent circuits and Floquet analysis to study the suggested structure with the principle of modulation. This work is divided into four parts: we start with an explication of the almost-periodic modal (or spectral) modulation and its spatial equivalent to examine strongly coupled cells [29–33]. Then, we applied MoM-GEC as a numerical method to solve the proposed problem [1–7,34–36]. Next, several numerical results are presented to confirm the validity of the approach. Finally, some conclusions are established.

## 2. Almost-Periodic Modulation to Study Strongly Coupled Arrays

The concept is a signal-processing concept for a filter with a periodic spectral response, as shown in [30–32]. Its response is described as an impulse response function that is given by:

$$\int_{-\infty}^{\infty} K(\Omega - \alpha)u(\alpha)\, d\alpha = V(\Omega) \tag{1}$$

Its Fourier representation of Equation (1) (in the spatial domain) yields to

$$H(x)U(x) = V(x) \tag{2}$$

From the transformations by way of analogy, which we took into account, we note that $x$ is a spatial coordinate and $\alpha \in [-\frac{\pi}{d_x}, \frac{\pi}{d_x}]$ is a spectral coordinate in the Brillouin domain and $d_x$ is a spatial period. Note that $\Omega$ is a spectral coordinate, as $\alpha$, where $H(x)$ is the Fourier transform of $K(\Omega)$ and is defined as the optical (or optoelectronic [30,31]) transfer function; $U(x)$ and $V(x)$ are the Fourier transforms of $u(\alpha)$ and $V(\Omega)$, respectively. More details are provided in [30].

### 2.1. Modulation in Infinite Almost-Periodic Arrays

Let us consider $f_\alpha(x)$ as a spectral periodic response for an infinite array that is written [4,33]:

$$f_\alpha(x) = \sum_{n=-N}^{N=+\infty} f_n(x)e^{+jn\alpha d_x} \tag{3}$$

$\alpha$ is a continuous Floquet mode $\alpha \in [-\frac{\pi}{d_x}, \frac{\pi}{d_x}]$; $x$ (or $x_n = x(n) = nd_x$) represents the position in the spatial domain (a spatial distribution) ; and $n$ is the position index in the periodic lattice.

with

$$f_n(x) = \frac{d}{2\pi} \int_{-\frac{\pi}{d}}^{\frac{\pi}{d}} f_\alpha(x)e^{-j\alpha nd_x}\, d\alpha \tag{4}$$

Considering $f_0$ is built from $x_0$, $f_1$ is identically constructed from $x_0 \pm d_x$. $f_0$ is a weight for $x_0$. In the same way, $f_1$ is a weight for $x_1 = x_0 \pm d_x$. Now, we can generalize the construction towards n elements. Then, $f_n(x) = f_0(x - nd_x)$, $n \in \mathbb{Z}$. $f_n(x)$ is a periodic function [34].

Now, let us put the given spectral modulation [30]:

$$U_{mod}^{\infty}(\alpha) = u(\alpha)f_\alpha(x) = u(\alpha)\sum_{n=-N}^{N=+\infty} f_n(x)e^{+j\alpha nd_x} \tag{5}$$

Next, we are considering: $TF(U_{mod}^{\infty}(\alpha)) = U_{mod}^{\infty}(x)$

It is possible to take $TF^{-1}(U_{mod}^{\infty}(\alpha)) = U_{mod}^{\infty}(x)$ (it depends the Fourier notation);
As a result,

$$U_{mod}^{\infty}(x) = \sum_{n=-N}^{N=+\infty} f_n U(x - nd_x) \tag{6}$$

A simple demonstration from (5) to (6) is provided:

$$\begin{aligned}
U_{mod}^{\infty}(x) &= TF^{-1}(U_{mod}^{\infty}(\alpha)) \\
&= TF^{-1}(u(\alpha)\sum_{n=-N}^{N=+\infty} f_n e^{+j\alpha x}) \\
&= \sum_{n=-N}^{N=+\infty} f_n TF^{-1}(u(\alpha)e^{+j\alpha x}) \\
&= \sum_{n=-N}^{N=+\infty} f_n(\frac{d_x}{2\pi}\int_{-\frac{\pi}{d_x}}^{\frac{\pi}{d_x}} u(\alpha)e^{+j\alpha x}e^{-j\alpha nd_x}\, d\alpha)
\end{aligned} \tag{7}$$

$$= \sum_{n=-N}^{N=+\infty} f_n \left( \frac{d_x}{2\pi} \int_{-\frac{\pi}{d_x}}^{\frac{\pi}{d_x}} u(\alpha) e^{+j\alpha(x-nd_x)} \, d\alpha \right)$$

$$= \sum_{n=-N}^{N=+\infty} f_n U(x - nd_x)$$

Equation (7) is established referring to the theorem of aliasing given in [37] and Equation (6.107) of [38].

Finally, a spatial modulation is derived from Equation (2) [24,30]

$$V_{mod}^\infty(x) = H(x) U_{mod}^\infty(x) = H(x) \sum_{n=-N}^{N=+\infty} f_n U(x - nd_x) \tag{8}$$

Notice that our old published work [34] on rectangular pulse functions can be a special case of this process of analysis and development, where $f_{n,\alpha} = \frac{W}{2d} \text{sinc}\left( \left( \frac{2n\pi}{d} + \alpha \right) \frac{w}{2} \right)$ are the Fourier series coefficients of the periodic pulse train (in the presence of Floquet modes), as described in [39].

For more details, following the pulse (or impulse) trains, we can introduce the Floquet phases as follows [4,33]:

$$f_\alpha(x) = \sum_{n=-N}^{N=+\infty} rect(x - nd_x) e^{+j\alpha x} \tag{9}$$

$$= \sum_{n=-N}^{N=+\infty} rect_n(x) e^{+j\alpha x}$$

Note that: $\sum_{n=-N}^{N=+\infty} rect_n(x) = \sum_{n=-N}^{N=+\infty} f_n e^{+j\frac{2n\pi}{d_x}x}$ (for a standard rectangular pulse train), which explains how to recover (reconstruct) a pulse train by means the Fourier series, as shown in Figure 1 and explained in [39,40] (see the subsection on periodic pulse and impulse trains in [39]). By adding the Floquet contribution $e^{+j\alpha x}$, we obtain the definition $f_\alpha(x) = \sum_{n=-N}^{N=+\infty} rect(x - nd_x) e^{+j\alpha x}$ (see Figure 1).

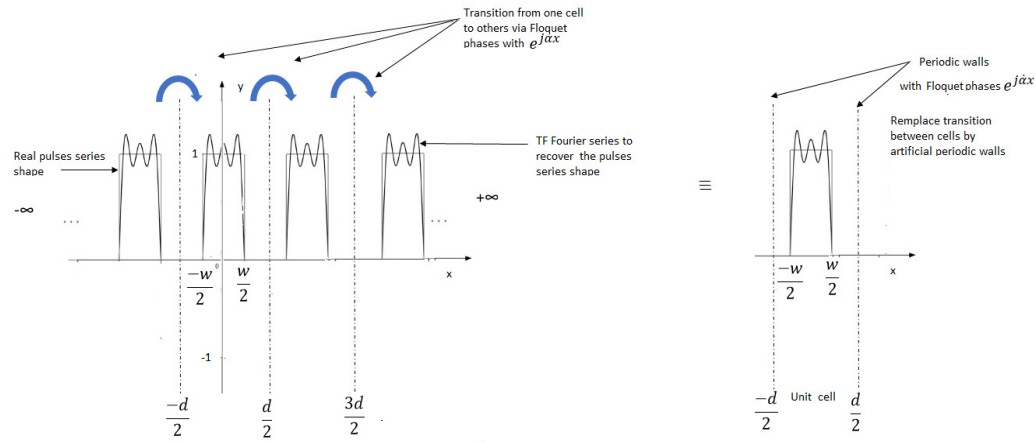

**Figure 1.** Construction of $f_\alpha(x)$ as given in Equations (9)–(11).

Rect is a rectangular function with a width W.

Then, we can develop a series of rectangle functions into a series of Fourier functions, which allows us to write: (see Figure 1 and [34])

$$f_\alpha(x) = \sum_{n=-N}^{N=+\infty} f_{n,\alpha} e^{+j\frac{2n\pi}{d_x}x} e^{+j\alpha x} = \sum_{n=-N}^{N=+\infty} f_{n,\alpha} e^{+j\left( \frac{2n\pi}{d_x} + \alpha \right)x} \tag{10}$$

we can note: $K_{x,n} = \frac{2n\pi}{d_x} + \alpha$ as a wavenumber that leads to decompose

$$f_\alpha(x) = \sum_{n=-N}^{N=+\infty} f_{n,\alpha} e^{+jK_{x,n}x} \tag{11}$$

With

$$
\begin{aligned}
f_{n,\alpha} &= \frac{1}{d_x} \int_{-\frac{d}{2}}^{\frac{d}{2}} f_\alpha(x) e^{-jK_{x,n}x} \, dx \\
&= \frac{1}{d_x} \int_{-\frac{w}{2}}^{\frac{w}{2}} e^{-jK_{x,n}x} \, dx \\
= \frac{W}{2d_x} \operatorname{sinc}(K_{x,n} \tfrac{w}{2}) &= \frac{W}{2d_x} \operatorname{sinc}((\tfrac{2n\pi}{d_x} + \alpha) \tfrac{w}{2})
\end{aligned}
\tag{12}
$$

Equation (12) is proven based on the example (Example 3.17) of [39].

Now let us apply the superposition theorem of Equation (10) (based on Floquet states) to generate the spatial solution [9,19]: it is also called modulation (see Equation (7))

$$
\begin{aligned}
\frac{d_x}{2\pi} \int_{-\frac{\pi}{d_x}}^{\frac{\pi}{d_x}} f_\alpha(x) e^{-j\alpha x} \, d\alpha &= TF^{-1}(f_\alpha(x)) \\
&= \frac{d_x}{2\pi} \int_{-\frac{\pi}{d_x}}^{\frac{\pi}{d_x}} (\sum_{n=-N}^{N=+\infty} f_{n,\alpha} e^{+j(\frac{2n\pi}{d_x}+\alpha)x}) e^{-j\alpha x} \, d\alpha \\
&= \frac{d_x}{2\pi} \int_{-\frac{\pi}{d_x}}^{\frac{\pi}{d_x}} (\sum_{n=-N}^{N=+\infty} f_{n,\alpha} e^{+j(\frac{2n\pi}{d_x})x}) \, d\alpha \\
&= \frac{d_x}{2\pi} \int_{-\frac{\pi}{d_x}}^{\frac{\pi}{d_x}} \sum_{n=-N}^{N=+\infty} rect_n(x) \, d\alpha \\
&= \frac{d_x}{2\pi} \int_{-\frac{\pi}{d_x}}^{\frac{\pi}{d_x}} \sum_{n=-N}^{N=+\infty} rect(x - nd_x) \, d\alpha \\
&= \frac{d_x}{2\pi} \sum_{n=-N}^{N=+\infty} rect(x - nd_x) \int_{-\frac{\pi}{d_x}}^{\frac{\pi}{d_x}} \, d\alpha \\
&= \sum_{n=-N}^{N=+\infty} rect(x - nd) \\
&= \sum_{n=-N}^{N=+\infty} rect(x - nd_x)
\end{aligned}
\tag{13}
$$

The spatial solution is a periodic pulses series
and is similar to $= U_{mod}^\infty(x)$ when$(u(\alpha) = 1)$

What we get in (13) is similar to Equation (14) of our published work [34] (and Equation (4) of the WATANABE reference [24]). Additionally, it is of the same type as Equation (6) and the expansion that follows in Equation (7). Then, a spatial modulation that was performed in Equation (8) follows.

### 2.2. Modulation in Finite Almost-Periodic Arrays

As previously explained in [4,6], the interactions between cells in the spectral domain for periodic finite arrays are governed by a discrete phase law such that $\alpha_p = \frac{2\pi p d_x}{D} = \frac{2\pi p}{N_x}$ (with $-\frac{N_x}{2} \le p \le -\frac{N_x}{2} - 1$), which comes from a rule-of-three math reasoning.

For a large period of finite arrays, $D \longrightarrow 2\pi$ ($2\pi$ is the hole interval of phases).

For a local period of finite arrays, $pd_x \longrightarrow \alpha_p =?$ (is the spectral contribution for one cell in position $pd_x$) ($p$ is the index position in a finite array, and $d$ is the local period).

So, $\implies \alpha_p = \frac{2\pi p}{N_x}$, where $N_x$ is the total number of elements in finite array, and $p$ is the index position [13].

Figure 2 explains how to discretize the phases from the infinite case to the finite case. According to the same Figure 2, each cell interacts spectrally with its neighbors through the continuous Floquet modes $\alpha$ (or the phase shift $e^{+j\alpha x}$) in the infinite case and $\alpha_p$ (or the phase shift $e^{+j\alpha_p x}$) in the finite case.

This allows writing the spectral solution as the sum of discrete Floquet states [4,33],

$$f_{\alpha_p}(x) = \frac{1}{\sqrt{N_x}} \sum_{n=-\frac{N_x}{2}}^{\frac{N_x}{2}-1} f_n(x) e^{+j\alpha_p n d_x} \tag{14}$$

and

$$f_n(x) = \frac{1}{\sqrt{N_x}} \sum_{p=-\frac{N_x}{2}}^{\frac{N_x}{2}-1} f_{\alpha_p}(x) e^{-j\alpha_p n d_x} \tag{15}$$

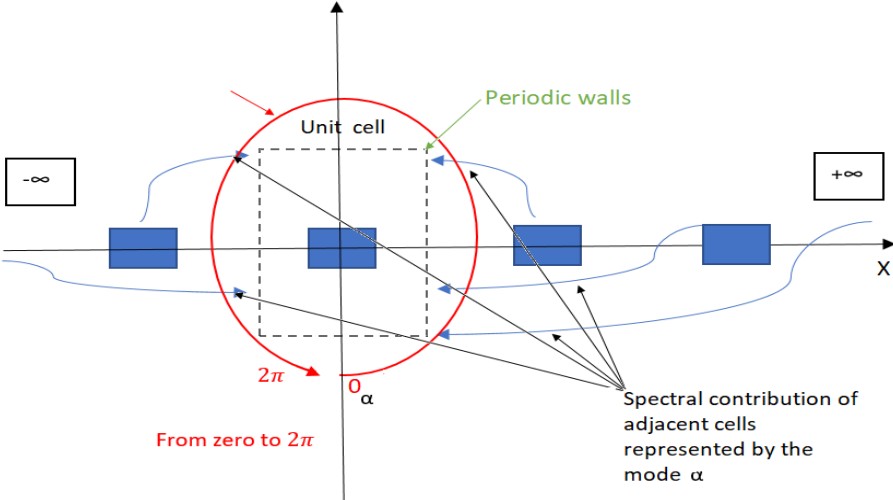

Contribution of continuous Floquet modes $\alpha$ in an infinite array

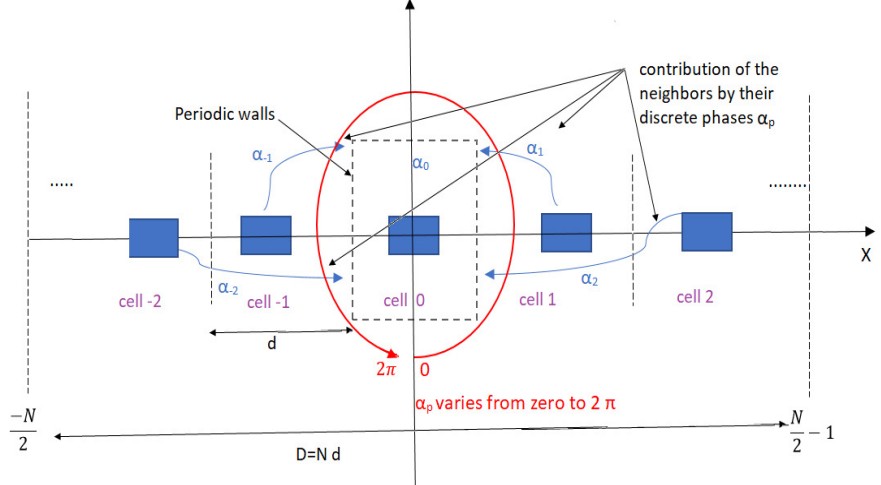

Contribution of discrete Floquet modes $\alpha_p$ in a finite array.

**Figure 2.** Spectral representation of the interactions of a unit cell with its neighbors (infinite and finite cases) (valid for strong coupling interaction by using Floquet phases).

In the same way, $f_n(x) = f_0(x - nd_x)$ with $-\frac{N_x}{2} \le n \le -\frac{N_x}{2} - 1$, and we can rewrite the spectral modulation law for a discrete Floquet mode [30],

$$U_{mod}^{Finite}(\alpha_p) = u(\alpha_p) f_{\alpha_p}(x) \tag{16}$$

$$= u(\alpha_p) \frac{1}{\sqrt{N_x}} \sum_{n=-\frac{N_x}{2}}^{\frac{N_x}{2}-1} f_n(x) e^{+j\alpha_p n d_x}$$

Thus, the DFT is written as $DFT(U_{mod}^{Finite}(\alpha_p)) = U_{mod}^{Finite}(x_i)$ with $x_i = id_x$ and $-\frac{N_x}{2} \le i \le -\frac{N_x}{2} - 1$

from which

$$U_{mod}^{Finite}(x) = \frac{1}{\sqrt{N_x}} \sum_{n=-\frac{N_x}{2}}^{\frac{N_x}{2}-1} f_n U(x - nd_x) \tag{17}$$

Eventually

$$V_{mod}^{Finite}(x) = H(x)U_{mod}^{Finite}(x) \tag{18}$$

$$= H(x)\frac{1}{\sqrt{N_x}} \sum_{n=-\frac{N_x}{2}}^{\frac{N_x}{2}-1} f_n U(x - nd_x)$$

## 3. MoM-GeC Modelization Based on Floquet Analysis

The problem formulation is already explained in [1–7]. It explains the development of the method of moments combined with Floquet spectral analysis to determine the electromagnetic performance in the presence of strong mutual couplings such as input impedance, surface current, surface electric field, radiated field, and directivity... etc.

## 4. Numerical Results

A part of our results was presented in [4,6,34,35]. Let us now display the other obtained results.

This approach can be applied to frequency-modulated continuous-wave (FMCW) radar antennas (to scan the radiation beam produced by very small areas of an antenna system) as well as to antennas that are massively placed in a coupled almost-periodic antenna array. The provided antenna example can be used to show how to model a 77 GHz (2 × 4) antenna array for frequency-modulated continuous-wave (FMCW) radar applications. The availability of antennas and antenna arrays in and on vehicles has become ordinary with the inclusion of remote crash-recognition-and-aversion systems, as well as lane-departure alerting systems. The two frequency bands appropriate for these systems are approximately 24 GHz and 77 GHz, respectively. In this model, we consider the microstrip patch antenna as a phased-array radiator. The dielectric substrate is air. According to Figures 3 and 4, the patch antenna has its first resonance (parallel resonance) at 24.52 and 77.9 GHz (after adjusting the length of the used antenna) . It is a common practice to shift this resonance to 24 and 77 GHz by scaling the length of the planar dipole antenna, as described in [35].

The next stage is to reconfirm the reflection coefficient of the planar antenna dipole, as depicted in Figure 5 and Figure 1 of [34]. The purpose of this check is to consider a good impedance match. It is very common to highlight the value as a limit value for the calculation of the bandwidth of the antenna. The deepest minima at 24 GHz and 77 GHz indicated a good fit with $120\pi$. The bandwidths of the antennas are roughly 1 GHz and 2 GHz, respectively. Thus, the spectrum bands are 23.5 GHz to 25.5 GHz and 76.5 GHz to 77.5 GHz. Finally, in terms of input impedances and S parameters, a good comparison between the adopted MoM GeC method and the references [41,42] is obtained.

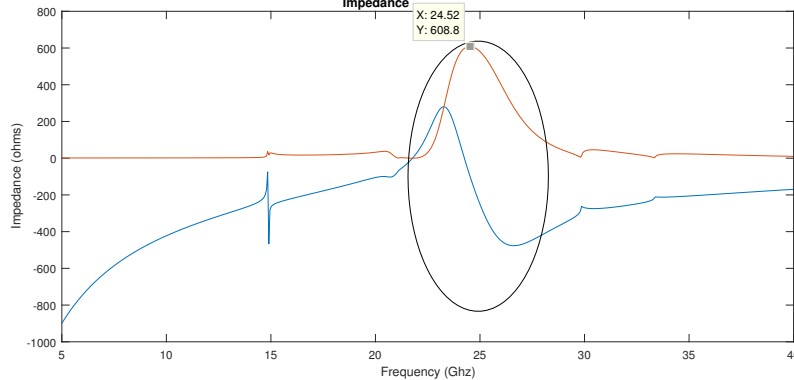

**Figure 3.** Impedance variation against frequency band around 24 GHz: obtained by the MoM-GEC method.

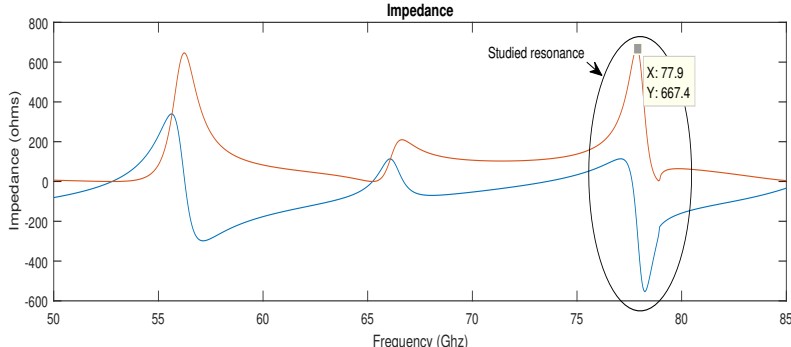

**Figure 4.** Impedance variation against frequency band around 77 GHz: obtained by the MoM GEC method.

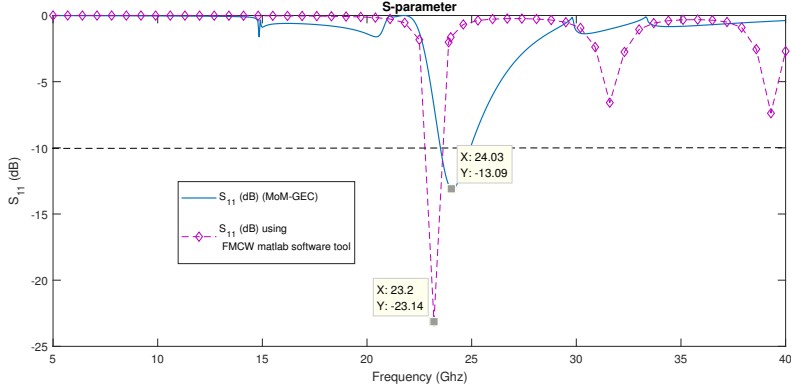

**Figure 5.** $S_{11}$-parameter (dB) variation against frequency band around 24 GHz: a comparison between the MoM-GEC and a MATLAB tool (see references [41,42]).

Therefore, the radiation pattern response for a small 2 × 4 antenna array is proposed based on the Floquet analysis (via the superposition theorem). Let us assume that the radar antenna system operates at 77 GHz with a bandwidth of 700 MHz. The following Figure 1 of [35] shows the spatial-radiation pattern of the resulting planar antenna using the superposition theorem of 2 × 4 Floquet radiation examples. Then, the sum of the discrete Floquet radiation patterns assigned to the FMCW radar permits the prediction of the global spatial-radiation pattern (what is called spatial modulation). A good comparison of the given numerical radiation pattern and the radiation obtained by patch array and cosine array is presented in Figure 3 of [35]. After validating the FMCW radar, we propose to evaluate the approach for a very large number of elements that uses the same frequency band at 24 and 77 GHz (for example, a lattice of 100 elements). Figure 6 gives an example of the superposition theorem (or a spatial modulation) for a large array to generate a spatial radiation pattern through the addition of the radiation patterns of Floquet states. After that, Figure 4 of [35] presents the variation of the spatial-radiation pattern (obtained using Floquet analysis) in the function of steering angles for 100 antenna elements that are distributed in a uni-dimensional configuration (for 5G application). In the same way, Figure 7 and Figure 5 of [35] show the variation of the 3D radiation pattern against different steering angles that are described with ($\theta_0 = 45°$, $\phi_0 = 0°$), and ($\theta_0 = 90°$, $\phi_0 = 0°$), in Cartesian coordinates and (u,v) space, respectively. Following the same study, Tables 1 and 2 show the directivity values for each Floquet state and the superposition theorem (in the two cases of the FMCW radar and the 5G application), with two different steering angles and at the frequencies of 24 GHz and 77 GHz. From Figure 1 of [35] and Figure 6, we can see that the radiation patterns are nearly identical and verify the condition where the directivities are similar for both the Floquet states and the superposition theorem. This is why, in both Tables 1 and 2, we find that all directivity values are identical (for the Floquet states as well as for the superposition), even when we change the steering angle. Knowing that generally,

the directivity value close to 20 dB satisfies a narrow beam angle of about 20 degrees, as shown graphically in Figure 7a.

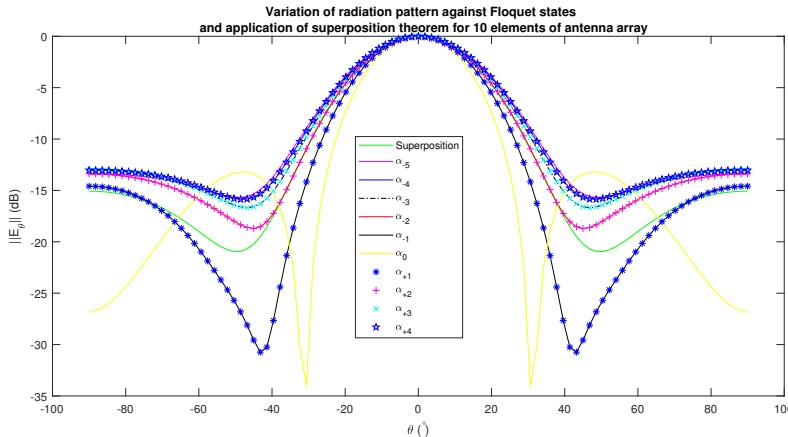

**Figure 6.** Variation of radiation pattern against Floquet states and application of superposition theorem for 10 elements of antenna array (uni-dimentionnal configuration) at 77 GHz: obtained by the MoM-GEC method.

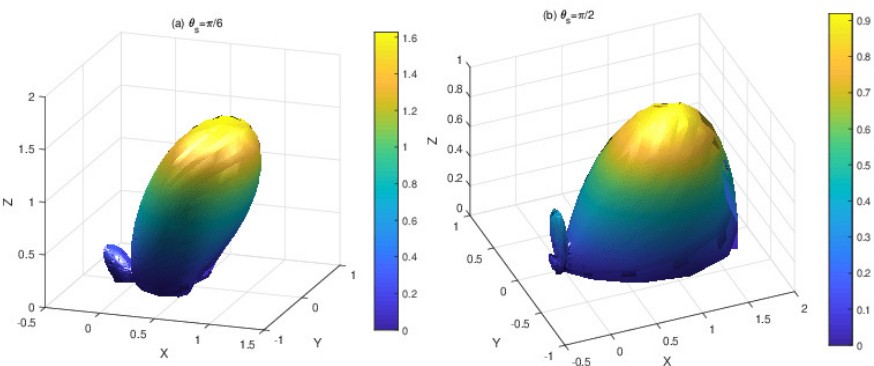

**Figure 7.** Variation of radiation pattern against Floquet states and application of superposition theorem for 100 elements of antenna array (uni-dimentionnal configuration) at 77 GHz: obtained by the MoM-GEC method.

**Table 1.** Directivity versus some Floquet states (considering 100 antenna arrays) and the superposition theorem (or the modulation as explained in Formula (13), which transformed to study a finite array) for $\phi_s = 0$, $\theta_s = 30°$ steering angles (used for 5G application).

| Floquet States | Directivity Values (dB) at 24 GHz |
|---|---|
| $(\alpha_{-49}, \beta = 0)$ | 23.0200 |
| $(\alpha_{-10}, \beta = 0)$ | 22.5953 |
| $(\alpha_{+20}, \beta = 0)$ | 22.8874 |
| $(\alpha_{+30}, \beta = 0)$ | 23.2055 |
| Superposition | 23.3009 |
| **Floquet States** | **Directivity Values (dB) at 77 GHz** |
| $(\alpha_{-49}, \beta = 0)$ | 22.8771 |
| $(\alpha_{-10}, \beta = 0)$ | 24.1083 |
| $(\alpha_{+20}, \beta = 0)$ | 23.4323 |
| $(\alpha_{+30}, \beta = 0)$ | 23.1353 |
| Superposition | 23.5783 |

**Table 2.** Directivity versus Floquet states and the superposition theorem (or the modulation as explained in Formula (13), which transformed to study a finite array) for $\phi_s = 0$, $\theta_s = 45°$ steering angles (FMCW radar application).

| Floquet States | Directivity Values (dB) at 24 GHz |
|---|---|
| $(\alpha_{-2}, \beta_{-1})$ | 21.7249 |
| $(\alpha_{-1}, \beta_{-1})$ | 26.7351 |
| $(\alpha_0, \beta_{-1})$ | 21.1290 |
| $(\alpha_{+1} \beta_{-1})$ | 20.8615 |
| $(\alpha_{-2}, \beta_0)$ | 20.6344 |
| $(\alpha_{-1}, \beta_0)$ | 21.0667 |
| $(\alpha_0, \beta_0)$ | 14.8990 |
| $(\alpha_{+1}, \beta_0)$ | 21.0737 |
| Superposition | 21.0455 |
| **Floquet States** | **Directivity Values (dB) at 77 GHz** |
| $(\alpha_{-2}, \beta_{-1})$ | 22.3762 |
| $(\alpha_{-1}, \beta_{-1})$ | 18.0573 |
| $(\alpha_0, \beta_{-1})$ | 21.9229 |
| $(\alpha_{+1} \beta_{-1})$ | 22.1662 |
| $(\alpha_{-2}, \beta_0)$ | 20.9345 |
| $(\alpha_{-1}, \beta_0)$ | 21.3088 |
| $(\alpha_0, \beta_0)$ | 13.7990 |
| $(\alpha_{+1}, \beta_0)$ | 21.3088 |
| Superposition | 21.7022 |

## 5. Conclusions

In this article, we illustrated the principle of Floquet spectral modulation based on the Fourier analysis (and its spatial form) to study almost-periodic sub-arrays (with finite size) in the presence of strong mutual coupling interaction, defined on infinite support (or a really large finite size). This study is very useful for the new generation of technologies based on millimeter and terahertz waves in phased arrays, for example, in dense-massive-MIMO, smart-surfaces, 5G, and 6G applications. In future work, we are interested to investigate the randomly modulated almost-periodic arrays (also in the presence of strong coupling).

**Author Contributions:** Conceptualization, H.B.; software, H.B. and A.T.; formal analysis, H.B. and A.T.; investigation, H.B. and A.T.; writing—original draft preparation, H.B. and A.T.; writing—review and editing, H.B. and A.T. All authors have read and agreed to the published version of the manuscript.

**Funding:** This project received funding from the Laboratory Sys'Com-ENIT (LR-99- ES21)-National Engineering School of Tunis ENIT, Tunis, Tunisia, 1002.

**Data Availability Statement:** Not applicable.

**Acknowledgments:** I would like to thank each of the Ammar Bouallegue (ENIT Tunis-El Manar University), Christophe Craeye (UCLouvain University), Junwu TAO (N7-University of Toulouse) and especiallyHenri Baudrand (N7-University of Toulouse), who died recently, for their help in the achievement of this work.

**Conflicts of Interest:** The authors declare no conflict of interest.

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
