# Peer review of "Floquet Spectral Almost-Periodic Modulation of Massive Finite and Infinite Strongly Coupled Arrays: Dense-Massive-MIMO, Intelligent-Surfaces, 5G, and 6G Applications"

_electronics, doi:10.3390/electronics11010036_

Round 1

Reviewer 1 Report

kindly find the attached review report.

Reviewer 2 Report

1. In Keywords: Replace "MM" for "mm", since it is refered to the millimeter waves;

2. "His response is described as an impulse..." : "His" should be replaced by "Its";

3. Equation 13: It has too many equations making the reading dense. Since it is a paper it should be easy to read. As such, I suggest presenting
three equations;

4. In Section 3: Briefly summarize what is explained in the referenced papers;

5. In Section 4: Indicate the results present in the referenced papers and explain them briefly;

6. As a sugestion, the results could have been obtained using CST Studio Suite.  
